# Bayesian model averaging for nonparametric discontinuity design

**Max Hinne** *, **David Leeftink, Marcel A. J. van Gerven, Luca Ambrogioni**

Donders Institute for Brain, Cognition and Behaviour, Radboud University, Nijmegen, The Netherlands

* max.hinne@donders.ru.nl

## Abstract

Quasi-experimental research designs, such as regression discontinuity and interrupted time series, allow for causal inference in the absence of a randomized controlled trial, at the cost of additional assumptions. In this paper, we provide a framework for discontinuity-based designs using Bayesian model averaging and Gaussian process regression, which we refer to as 'Bayesian nonparametric discontinuity design', or BNDD for short. BNDD addresses the two major shortcomings in most implementations of such designs: overconfidence due to implicit conditioning on the alleged effect, and model misspecification due to reliance on overly simplistic regression models. With the appropriate Gaussian process covariance function, our approach can detect discontinuities of any order, and in spectral features. We demonstrate the usage of BNDD in simulations, and apply the framework to determine the effect of running for political positions on longevity, of the effect of an alleged historical phantom border in the Netherlands on Dutch voting behaviour, and of Kundalini Yoga meditation on heart rate.

**Data Availability Statement:** All data and analyses are available on Github: https://github.com/mhinne/BNQD.

**Funding:** The author(s) received no specific funding for this work.

## Introduction

The bread and butter of scientific research is the randomized-controlled trial (RCT) [1]. In this design, the sample population is randomly divided into two groups; one that is manipulated (e.g. a drug is administered or a treatment is performed), while the other is left unchanged. RCT allows one to perform *causal inference*, and learn about the causal effect of the intervention [2, 3]. However, in practice there may be several insurmountable ethical or pragmatic hurdles that deter one from using RCT, such as ethical or pragmatic concerns. Luckily, all is not lost for experimental design. There exist several *quasi*-experimental designs (QEDs) that replace random assignment with deterministic assignment, which still allow for causal inferences, but at the cost of additional assumptions [4]. Prominent examples are regression discontinuity (RD) and interrupted time series (ITS) designs, that assign a sample to one of the two groups based on it passing a threshold on an assignment variable [5–7]. The idea behind these approaches is that, around the assignment threshold, observations are distributed essentially randomly, so that locally the conditions of RCT are recreated [8, 9]. The methodological pipeline of quasi-experimental designs like these generally consists of three steps [10]. First, a regression (typically linear) is fit to each of the two groups individually.

**Competing interests:** The authors have declared that no competing interests exist.

Next, the regressions are extrapolated to the threshold (RD), or to the entire post-intervention range (ITS). Finally, the difference between the extrapolations of the two groups is taken as the effect size of the intervention. A straightforward statistical test can be applied to check whether the effect is present.

Here, we provide a novel framework for such approaches, which we call 'Bayesian nonparametric discontinuity design' (BNDD). The main innovations of BNDD are: First, we frame the problem of detecting an effect as Bayesian model comparison. Instead of comparing the pre- and post-intervention regressions, we introduce a continuous model and a discontinuous model. In the discontinuous model, observations before and after the intervention are assumed to be independent, while in the continuous model this assumption is lifted. We quantify the evidence in favor of either model, rather than only for the alternative model, via Bayesian model comparison [11]. This enables the computation of the marginal effect size via Bayesian model averaging, which provides a more nuanced estimate compared to implicitly conditioning on the alternative model [12, 13]. Furthermore, the model comparison approach automatically penalizes the discontinuous model for its additional flexibility [14]. Second, we use Gaussian process (GP) regression to avoid strong parametric assumptions. The result is a flexible model that can capture nonlinear interactions between the predictor and outcome variables. Traditional assumptions, such as linearity, can still be implemented in our model by using the appropriate covariance function. At the same time, much more expressive covariance functions can be used, such as the spectral mixture kernel [15], that better capture long-range correlations, and lead to more accurate inference. Lastly, in most discontinuity-based methods for quasi-experimental design, a bandwidth parameter determines the trade-off between estimation reliability and the local randomness assumptions that are needed to draw causal inferences [16]. In BNDD, all observations are used to estimate both the continuous and discontinuous model, but by optimizing the length-scale parameter of the GP covariance functions we control the sensitivity to different types of discontinuities and adherence to locality assumptions.

## Related work

While quasi-experimental designs have been around since the 1960s [5, 17], recently there has been a renewed interest in this class of methods [9, 18], in particular in epidemiology [19] and education [20]. Researchers from different domains are promoting the use of QED [21–23], which has prompted several extensions of classical QEDs. For instance, several authors have proposed to use Bayesian models for QED [16, 24]. By assuming a prior distribution for alleged effect size and using Bayes' theorem, these studies provide an explicit descriptions of the estimation uncertainty. In contrast to our work, these methods focus on the estimation of the treatment effect instead of model comparison, and typically assume restrictive parametric forms. Other studies have considered nonparametric alternatives to linear models. For example, [25] use locally linear nonparametric regression. Alternatively, one can use kernel methods that compute a smoothly weighted average of the data points to create an interpolated regression that does not depend on a specific parametric form [20]. Other studies have considerd using Gaussian process regression for regression discontinuity as well [10, 26]. Here, instead of fitting a parametric form such as linear regression, the regression is modelled by a GP, which results in a flexible, nonparametric model and more accurate effect size estimates compared to when using linear regression. BNDD uses GP regression as well, but whereas [10, 26] focus on the inference of the magnitude of the treatment effect, we first determine whether an effect is present at all using Bayesian model comparison [11], and we use Bayesian model averaging [12] to reduce the overconfidence that follows from conditioning on the alternative

model or particular covariance functions. Consequently, BNDD is less prone to false positives, is able to detect discontinuities in derivatives of the latent function rather than in the function per sé, and using the spectral mixture kernel our approach is well-suited for detecting changes in time series, which is crucial in ITS design.

## Discontinuity-based causal inference

We provide a brief introduction of the background of causal inference using RD and ITS designs, for a more in-depth discussion, we refer to e.g. [7, 27]. The detection of a causal effect is naturally formulated using the potential outcomes framework [28], which assumes that for each individual in the study both the outcome of the treatment and its alternative can potentially be observed.

Consider an observation $i$ (with or without temporal ordering) with independent variable $x_i \in \mathbb{R}^p$ and response $y_i \in \mathbb{R}^q$ (we will assume $p = q = 1$, but multidimensional extensions are straightforward). In addition, we observe an indicator variable $z_i$, where $z_i = 1$ denotes the intervention of interest has been applied to case $i$, and $z_i = 0$ indicates it has not. The outcome depends on treatment, so

$$y_i = \begin{cases} y_i(0) & \text{if } z_i = 0, \\ y_i(1) & \text{if } z_i = 1. \end{cases} \tag{1}$$

The individual causal effect is defined as the difference between these two potential outcomes, that is $d_i = y_i(1) - y_i(0)$. Since we only ever observe one outcome, the individual causal effect is out of reach, so in RD design we focus on the *average causal effect* (ACE) instead, defined by the differences in the expectations:

$$d_{\text{ACE}} = \mathbb{E}[y(1)] - \mathbb{E}[y(0)] \ . \tag{2}$$

In the randomized controlled trial, the assignment of treatment $z_i$ is random, so that all differences other than due to the treatment are integrated out in these expectations [20]. In QED designs such as RD and ITS however, the allocation to intervention or control group is based on a threshold $x_0$ [29]:

$$z_i = \begin{cases} 1 & \text{if } x_i \geq x_0 \text{ and} \\ 0 & \text{otherwise.} \end{cases} \tag{3}$$

This changes how the ACE is computed, which for RD design becomes [8, 30]:

$$\begin{aligned} d_{\text{RD}} &= \mathbb{E}[y_i(1) - y_i(0) \mid x_i = x_0] \\ &= \lim_{x \downarrow x_0} \mathbb{E}[y_i \mid x_i = x_0] - \lim_{x \uparrow x_0} \mathbb{E}[y_i \mid x_i = x_0] \ , \end{aligned} \tag{4}$$

provided the distributions of $y_i$ given $x_i$ are continuous in $x$, and the conditional expectations $\mathbb{E}[y_i(1) \mid x_i]$ and $\mathbb{E}[y_i(0) \mid x_i]$ exist.

For interrupted time series, there are no post-intervention control observations, as all post-threshold observations $x_i \geq x_0$ are in the intervention group. Here, the causal estimand becomes the *average effect of the treatment on the treated* (ATT) [31]:

$$\begin{aligned} d_{\text{ITS}}(x_i) &= \mathbb{E}[y_i(1) - y_i(0) \mid x_i \geq x_0] \\ &= \mathbb{E}[y_i \mid x_i, D] - \mathbb{E}[y_i \mid x_i, D^0] \ , \end{aligned} \tag{5}$$

for $x_i \geq x_0$, $D = \{(x_i, y_i)\}_{i=1}^n$, and $D^0 = \{(x_i, y_i)\}_{x_i < x_0}$. Intuitively, this measure of effect size is

the difference between the extrapolation based on the pre-intervention data, and the actual post-intervention observations. Due the the reliance on extrapolation, it is crucial that correct assumptions are made on the functional form. For example, assuming linearity will lead to a biased ATT estimate if this does not describe the functional form well.

Importantly, for both approaches we assume there are no confounding variables that affect the relationship between $x$ and $y$ (for a more in-depth discussion of RD design, see e.g. [16]).

## Bayesian nonparametric discontinuity design

In standard RD and ITS analyses, causal conclusions are drawn by estimating the effect $d$ and testing whether this differs from zero. Instead, we perform Bayesian model comparison to see whether the data are better supported by the alternative model $\mathcal{M}_1$, that claims an effect is present, than by the null model $\mathcal{M}_0$, in which such an effect is absent. The result of the model comparison is quantified by the Bayes factor [32]:

$$\mathrm{BF}_{10} = \frac{p(D \mid \mathcal{M}_1)}{p(D \mid \mathcal{M}_0)} \quad . \tag{6}$$

Here, $p(D \mid \mathcal{M}_1)$ and $p(D \mid \mathcal{M}_0)$ are the marginal likelihoods of the two models with their respective parameters integrated out. The Bayes factor indicates how much more likely the data are given the discontinuous model, compared to the continuous model [33]. Unlike a $p$-value, it can provide evidence for either model, so that it is possible to find evidence supporting the *absence* of a discontinuity [11, 34]. Furthermore, this model comparison approach automatically accounts for model complexity [14].

In the null model, all probability mass of $p(d \mid D, \mathcal{M}_0)$ is concentrated at $d = 0$, while for the alternative model we have an effect size distribution $p(d \mid D, \mathcal{M}_1)$. Existing regression discontinuity methods focus on inference of $d$, and hence implicitly condition on $\mathcal{M}_1$. This approach ignores the uncertainty in the model posterior

$$p(\mathcal{M} \mid D) = \frac{p(D \mid \mathcal{M})p(\mathcal{M})}{\sum_i p(D \mid \mathcal{M}_i)p(\mathcal{M}_i)} \quad , \tag{7}$$

where $p(\mathcal{M}_i)$ is the prior probability of model $i$. Ignoring the uncertainty in this distribution results in an overconfident overestimate of the effect size, and consequently of too optimistic conclusions of the efficacy of an intervention. This uncertainty can be accounted for via the Bayesian model average (BMA) estimate of $d$:

$$p(d \mid D) = \sum_{j=0,1} p(d \mid D, \mathcal{M}_j)p(\mathcal{M}_j \mid D) \quad . \tag{8}$$

The resulting distribution integrates over the uncertainty of the model, which has been shown to lead to optimal predictive performance [12]. Since the effect size is by definition zero according to $\mathcal{M}_0$, Eq (8) is a spike-and-slab distribution that combines a spike at $d = 0$ with a Gaussian distribution determined by $\mathcal{M}_1$, where each component is weighted by the posterior probability of the corresponding model. Compared to the overconfident estimation of $d$ conditioned only on $\mathcal{M}_1$, this has a regularizing effect [35], shrinking small effect size estimates towards zero. Note that for now, we assume a uniform prior over the models, such that $p(\mathcal{M}_0) = p(\mathcal{M}_1) = 1/2$, but this may be changed, for instance to account for multiple comparisons [36]. We proceed to explain the distributions implied by the two models in more detail.

## The continuous model

The continuous (null) model $\mathcal{M}_0$ implies that the regression does not depend on the threshold, which leaves us with a single regression for all data points. We assume Gaussian observation noise:

$$y_i \sim \mathcal{N}(f_0(x_i), \sigma_n^2) \ .$$

Here, $\sigma_n^2$ is the observation noise variance, and $f_0(x_i)$ captures the relationship between the predictor and the response. We do not impose a parametric form on $f_0$, and instead assume $f_0$ follows a Gaussian process (GP) [37]:

$$f_0 \mid \mathcal{M}_0 \sim \mathcal{GP}(\mu(x; \theta_0), k(x, x'; \theta_0)) \ ,$$

with mean function $\mu(x;\theta_0)$ and covariance function $k(x, x';\theta_0))$. We omit the dependence on the hyperparameters $\theta$ when confusion is unlikely to arise.

## The discontinuous model

In the alternative model we assume the latent processes before and after $x_0$ are independent. We write

$$f_1 \mid \mathcal{M}_1 \sim \mathcal{GP}(\mu(x; \theta_1), k_1(x, x'; \theta_1) \ , \tag{9}$$

where $k_1(x, x';\theta_1) = k(x, x';\theta_1)$ if $x$ and $x'$ are on the same side of $x_0$, and $k_1(x, x';\theta_1) = 0$ otherwise. As a result, the Gram matrix with elements $\mathbf{K}_{ij} = k_1(x_i, x_j;\theta_1)$ is block-diagonal:

$$\mathbf{K} = \begin{bmatrix} \mathbf{A} & 0 \\ 0 & \mathbf{B} \end{bmatrix} \ , \tag{10}$$

with the elements in the matrices $\mathbf{A}$ and $\mathbf{B}$ corresponding to the covariances between observations at the same side of the threshold $x_0$. For computational efficiency, the inverse of $\mathbf{K}$ can be computed by the separate inverses of these smaller sub-matrices.

## Regression discontinuity effect size

Since $f_1$ is continuous everywhere except at $x_0$, we can determine the effect size given $\mathcal{M}_1$ by taking the difference of its limits as in Eq (4). The result is a Gaussian distribution:

$$p(d \mid D, \mathcal{M}_1) = \mathcal{N}(m, s^2) \ , \tag{11}$$

with

$$m = \lim_{x \downarrow x_0} f_1(x) - \lim_{x \uparrow x_0} f_1(x) \tag{12}$$

and

$$s^2 = \lim_{x \downarrow x_0} \mathbb{V}[f_1(x)] + \lim_{x \uparrow x_0} \mathbb{V}[f_1(x)] = 2\sigma_n^2 \ , \tag{13}$$

for stationary covariance functions, where $\sigma_n^2 \in \theta_1$ represents the observation noise hyperparameter of the discontinuous model.

## Interrupted time series effect size

In contrast to RD design, in ITS the discontinuity may induce a nonstationarity in the latent process, such as a change in length-scale or frequency. To address this, we allow the

hyperparameters pre- and post-intervention to differ, i.e. $\mathbf{A}_{ij} = k(x_i, x_j; \theta_1^{\mathbf{A}})$ and $\mathbf{B}_{ij} = k(x_i, x_j; \theta_1^{\mathbf{B}})$. The differences in design also imply a different notion of effect size, which is now a function of $x$:

$$p(d(x) \mid D, \mathcal{M}_1) = \mathcal{N}(m(x), s_n^2) \ , \tag{14}$$

with $m(x) = f_1(x; \theta_1^{\mathbf{B}}) - f_1(x; \theta_1^{\mathbf{A}})$ and $s_n^2 = (\sigma_n^{\mathbf{A}})^2 + (\sigma_n^{\mathbf{B}})^2$. Note that $s_n^2$ does not depend on $x$. In practice, we summarize this dynamic effect by its maximum.

Of particular interest in ITS are covariance functions that capture long-range correlations, because these have the potential to extrapolate better and hence provide more accurate effect size estimates. The spectral mixture kernel was designed for this purpose [15]. It is defined as a mixture of Gaussian components in the frequency domain:

$$S(\omega) = \sum_{q=1}^{Q} w_q \frac{1}{\sigma_q \sqrt{2\pi}} \exp\left[ -\frac{1}{2} \left( \frac{\omega - \mu_q}{\sigma_q} \right)^2 \right] \ , \tag{15}$$

where $\mu_q$ and $\sigma_q^2$ are the mean and variance of each component, respectively. This spectral representation is then transformed into a regular stationary covariance function using the inverse Fourier transform [38], which results in

$$k(\tau) = \sum_{q=1}^{Q} w_q \cos(2\pi\tau\mu_q)\exp(-2\pi^2\tau^2\sigma_q^2) \ , \tag{16}$$

with $\tau = |x - x'|$. The hyperparameters $\theta = (Q, \mu, \sigma, \mathbf{w})$ have the following meaning: $Q$ is the number of mixture components, $\mu_q$ indicates the mean frequency of component $q$, the inverse of the variance $1/\sigma_q$ can be interpreted as the length-scale of each component, reflecting how quickly that frequency contribution changes with the input $x$, and the weights $w_q$ determine the relative contribution of each component [15].

## Model training

The marginal likelihood of Gaussian process regression with Gaussian observation noise is available in closed form [37], but unfortunately this is not the case for the model marginal likelihood that integrates over the hyperparameters $\theta$, which is needed to compute the Bayes factor. We therefore approximate these using the Bayesian Information Criterion (BIC) [39], given by as

$$\log p(D \mid \mathcal{M}_i) \approx \log p(\mathbf{y} \mid \mathbf{x}, \hat{\theta}, \mathcal{M}_i) - \frac{l}{2}\log n \ , \tag{19}$$

with $\mathbf{x} = (x_1, \ldots, x_n)^T$ and $\mathbf{y} = (y_1, \ldots, y_n)^T$, $l$ the number of hyperparameters, and $\hat{\theta} = \arg\max_{\theta} p(\mathbf{y} \mid \mathbf{x}, \theta, \mathcal{M})$ the optimized hyperparameters, for $i \in \{0, 1\}$.

BNDD is implemented in Python using GPflow 2.2 [40]. We set the prior function to the empirical mean. The BMA distribution is approximated via Monte Carlo, and visualized with kernel density estimation. Code and data are available at Github.

**Training the spectral mixture kernel.** The number of mixture components $Q$ in the spectral mixture kernel of our ITS approach is optimized in the same way as other covariance function parameters are optimized, that is, by optimizing the GP marginal likelihood. The covariance function mixture parameters are initialized by fitting a Gaussian mixture model to the empirical spectral using the Lomb-Scargle periodogram, which is applicable for detecting spectral features in (potentially) unevenly sampled data [41].

## Covariance functions as design choices

The choice of the Gaussian process covariance functions plays two conceptually distinct roles in BNDD. First, our choice of covariance function reflects our beliefs about the latent process that generated the observations. In traditional RD designs, one assumes a parametrized model such as (local) linear regression. In BNDD, this explicit parametric form is replaced by a GP prior that assigns a probability distribution to the space of functions. For instance, we may expect functions to be smooth in $x$, or assume functions are a superposition of sine waves [15, 37]. BNDD can replicate parametrized models by selecting degenerate covariance functions, such as a linear covariance function.

These modeling choices are crucial in RD design as model misspecification can lead to incorrect inference. When we do not have clear prior beliefs about a covariance function, we may compute the Bayesian model average [12] across a set of candidate kernels $\mathcal{K}$:

$$\mathrm{BF}_{10}^{\mathrm{total}} = \frac{p(D \mid \mathcal{M}_1)}{p(D \mid \mathcal{M}_0)} = \frac{\sum_{k \in \mathcal{K}} p(D \mid k) p(k \mid \mathcal{M}_1)}{\sum_{k \in \mathcal{K}} p(D \mid k) p(k \mid \mathcal{M}_0)} \quad . \tag{20}$$

Here, the quantity $\mathrm{BF}_{10}^{\mathrm{total}}$ serves as a final decision metric to determine an effect in a quasi-experimental design, while a detailed report is provided by inspecting the Bayes factors corresponding to each considered covariance function. Similarly, we can compute a marginal effect size across all considered kernels. In practice, the evidence of one covariance function can dominate all others, in which case the BMA procedure converges to performing the analysis with the best covariance function only.

The second role of the covariance function choice is that it determines to which types of discontinuities BNDD is sensitive. Importantly, different covariance functions can be used to test fundamentally different hypotheses, as they determine which features of the latent function are part of the alleged effect. For example, the simplest (degenerate) covariance function, the constant function, is sensitive only to differences in the means of the two groups (resulting essentially in a quasi-experimental Bayesian t-test), while the linear covariance function is sensitive to both the difference in mean as well as in slope. In the non-degenerate case, the Matérn covariance function with parameter $v = p + 1/2$ can detect discontinuities in up to the $p$-th derivative. It has two interesting special cases: one is the exponential covariance function (Matérn with $p = 1/2$), which detects only discontinuities in the function itself (and not in its derivatives). This is the nonparametric counterpart of traditional linear regression discontinuity. On the other end is the exponentiated-quadratic covariance function which (Matérn kernel with $v = \infty$). This allows us to detect discontinuities of any order, although the amount of data required to detect such subtle effects may become prohibitively large.

## Simulations

We evaluate the performance of BNDD in regression discontinuity settings in simulations, using polynomials up to the fifth order, which have been used in other RD design studies as well [30]. We evaluate the performance of BNDD using the linear, exponential, Matérn ($v = 3/2$) and exponentiated-quadratic covariance functions and compare its results with two baselines. The first is the Python RDD package, which uses linear regression together with the Imbens-Kalyanaraman bandwidth selection method [30] to select only a subset of the data around $x_0$ to perform the analysis on. The second comparison is with another GP-based approach [26], which first estimates the conditional effect size distribution $p(d \mid \mathcal{M}_1, D)$ and then tests the null hypothesis $d = 0$. We refer to this approach as the 2-stage GP as it combines the GP regression from $\mathcal{M}_1$ with a frequentist test.

The true data generating functions used are provided in [30], and are complemented by a simple linear function to see the behaviour when the linearity assumption by the baseline is actually correct. The function definitions are given by

$$f_{\text{Linear}}(x) = 0.23 + 0.89x$$

$$f_{\text{Quad}}(x) = \begin{cases} 3x^2 & \text{if } x < x_0, \\ 4x^2 & \text{otherwise.} \end{cases}$$

$$f_{\text{Cubic}}(x) = \begin{cases} 3x^3 & \text{if } x < x_0, \\ 4x^3 & \text{otherwise.} \end{cases}$$

$$f_{\text{Lee}}(x) = \begin{cases} 0.48 + 1.27x + 7.18x^2 + 20.21x^3 + 21.54x^4 + 7.33x^5 & \text{if } x < x_0, \\ 0.48 + 0.84x - 3.0x^2 + 7.99x^3 - 9.01x^4 + 3.56x^5 & \text{otherwise.} \end{cases}$$

$$f_{\text{CATE1}}(x) = 0.42 + 0.84x - 3.0x^2 + 7.99x^3 - 9.01x^4 + 3.56x^5$$

$$f_{\text{CATE2}}(x) = 0.42 + 0.84x + 7.99x^3 - 9.01x^4 + 3.56x^5$$

$$f_{\text{Ludwig}}(x) = \begin{cases} 3.71 + 2.3x + 3.28x^2 + 1.45x^3 + 0.23x^4 + 0.03x^5 & \text{if } x < x_0, \\ 3.71 + 18.49x - 54.81x^2 + 74.3x^3 - 45.02x^4 + 9.83x^5 & \text{otherwise.} \end{cases}$$

$$f_{\text{Curvature}}(x) = \begin{cases} 0.48 + 1.27x - 3.44x^2 + 14.147x^3 + 23.694x^4 + 10.995x^5 & \text{if } x < x_0, \\ 0.48 + 0.84x - 0.3x^2 - 2.397x^3 - 0,901x^4 + 3.56x^5 & \text{otherwise.} \end{cases}$$

For each latent function $f$, we generate 100 data sets with $n = 100$ observations each $(x_i, y_i)$ according to the following procedure:

$$x_i \sim \mathcal{U}(-1, 1)$$
$$y_i \mid x_i, \sigma, d, f \sim \mathcal{N}(f(x_i) + d[x_i \geq x_0], \sigma^2),$$

where the threshold $x_0 = 0$. We fix $\sigma = 1.0$ and vary $d \in \{0, 0.5, \ldots, 4.0\}$, effectively providing a range of different signal-to-noise regimes.

Next, we subject the simulated data to analysis by BNDD, using a first-order polynomial, an exponential, a Matérn ($\nu = 3/2$) and a exponentiated-quadratic covariance function, as well as the Bayesian model average of this set. Fig 1 shows an example run of BNDD on the functions

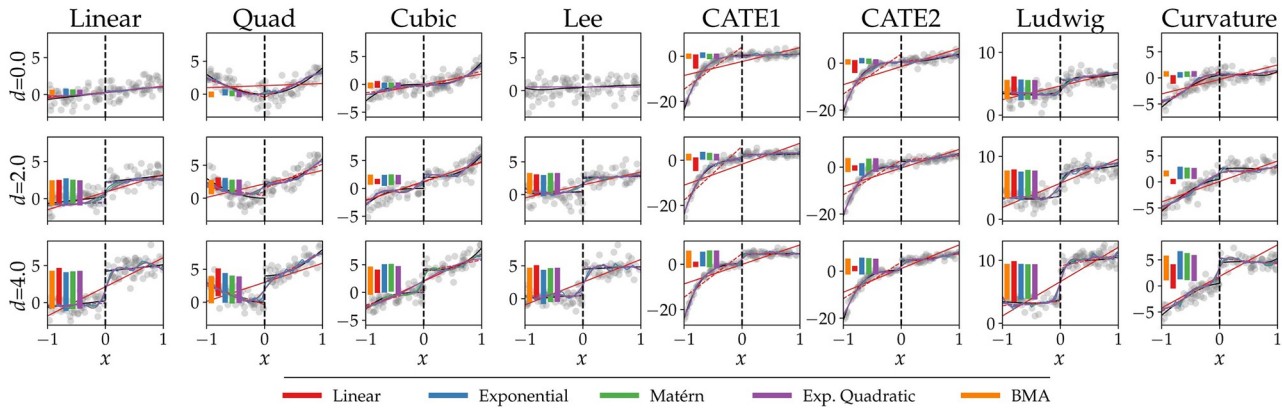

**Fig 1. Regression discontinuity example.** One simulation run for effect sizes $d \in \{0.25, 1.0, 4.0\}$ and $\sigma = 1.0$. The covariance functions used here are linear, exponential, Matérn ($\nu = 3/2$) and exponentiated-quadratic. The vertical bars indicate the estimated effect sizes by the discontinuous models for the different covariance functions. As the figure shows, the linear covariance function tends to have the strongest bias, in particular in the low signal-to-noise regime.

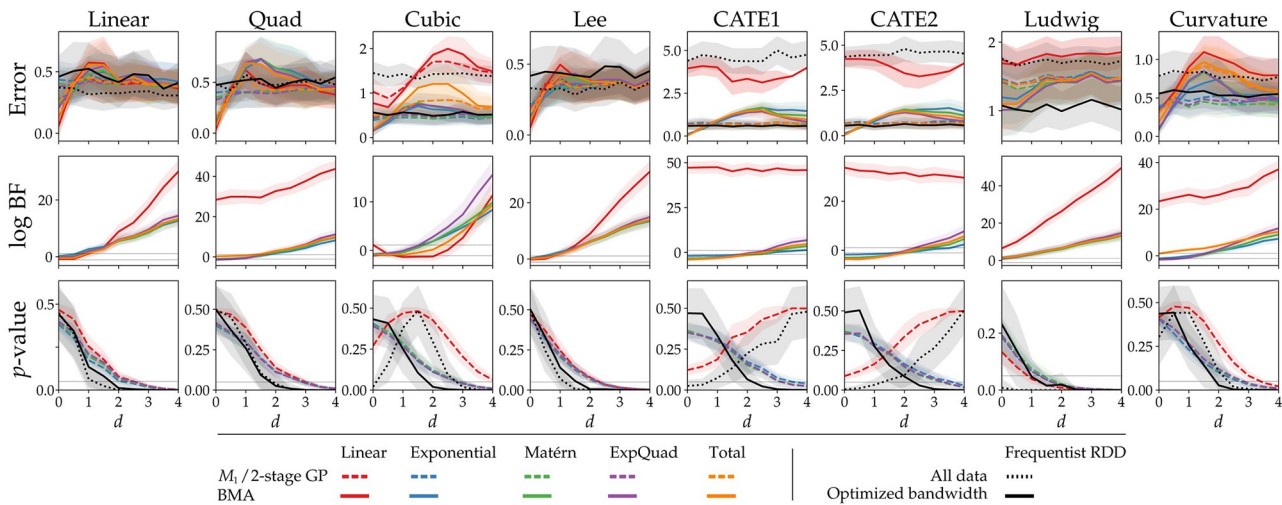

**Fig 2. Simulation results.** Top row: the error between the true and estimated effect size. The dashed line indicates the 2-stage GP approach (see text), which is equivalent to $\mathcal{M}_1$. Middle row: The log Bayes factor. Final row: The $p$-values obtained by the RDD baseline (black) and the 2-stage approach. The horizontal dashed lines indicates the common thresholds of $|BF|<3$ and $p = 0.05$. Error bars indicate standard errors over simulations runs.

considered in our simulation. Here, the different functions are shown together with the regressions by both the continuous and discontinuous models, for each of the four considered covariance functions. The vertical bars in the figure show the expectation of the estimated effect size $p(d \mid D, \mathcal{M}_1)$.

For each covariance function, we compute the Bayes factor for the presence of a discontinuity, and we estimate both the conditional and marginal effect size (8). For the RDD baseline with bandwidth optimization, and the 2-stage GP approach [26], we compute both the estimated effect size as well as the $p$-value for the test of a present effect. The performance of the different approaches is quantified by the absolute error between the estimated and true effect size. In addition, we show the decision metric for each method.

Fig 2 shows the absolute difference between the true effect size and the posterior expectations, as well as these decision metrics. The discontinuous model overestimates $d$ when the true effect size is small, as is to be expected from the implicit conditioning on an effect. The BMA does not have this bias, resulting in lower errors for small and absent effects. For medium effects, this itself can result in a bias due to shrinkage (e.g. the Cubic function), while for large effects the BMA converges to $\mathcal{M}_1$ and the bias disappears. Generally, BNDD performs on par with the optimized-bandwidth baseline, with worse performance for the Ludwig function, and better for e.g. Curvature, as well as for most cases with an absent or small effect.

The decision metrics show that for small or absent effects, BNDD can report evidence in favor of the null, while the corresponding $p$-values are inconclusive. The methods positively identify effects at roughly the same true effect sizes. An interesting special case is observed for the Lee and Ludwig functions, which both feature a discontinuity in their derivative [26], which is correctly picked up by BNDD even when the magnitude of the effect is small, confirming the ability to detect discontinuities of higher orders.

## Simulated ITS

We explore the ITS application of BNDD in another simulation. Here, we generate oscillating data where for $x \geq x_0$ a frequency shift is introduced. The latent function for the ITS

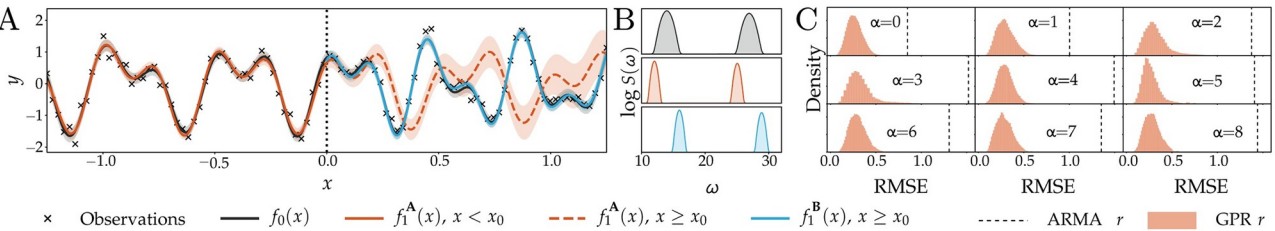

**Fig 3. ITS simulation. A**. ITS application. Model fit and extrapolation of $\mathcal{M}_0$ and $\mathcal{M}_1$. The data were generated with a post-intervention frequency shift of $\alpha = 4$. We find logBF = 0.15. The shaded interval represents two standard deviations around the mean. **B**. Estimated power spectra. The colors of the power density spectrum correspond to the legend of the regression. **C**. The RMSE between the estimated and true $d_{ITS}$ using posterior samples of BNDD and an ARMA baseline (dashed line).

simulation is given by

$$f(x) = \begin{cases} \sin(12x) + \dfrac{2}{3}\cos(25x) & \text{for } x < x_0 \text{ and} \\ \sin((12 + \alpha)x) + \dfrac{2}{3}\cos((25 + \alpha)x) & \text{for } x \geq x_0, \end{cases}$$

with $x_0 = 0$, and where $\alpha$ indicates the shift in frequency (set to $\alpha = 4$ in the example figure). We vary $\alpha$ across the range $[0, \ldots, 8]$ Hz. For observation noise, we once more assume

$$y \sim \mathcal{N}(f(x), \sigma^2) \ ,$$

and $\sigma^2 = 0.2$. For each value of $\alpha$, we generate 20 datasets containing $n = 200$ evenly spaced observations.

We compare our extrapolations based on the spectral mixture covariance function with an ARMA model, which is commonly used in ITS designs [42, 43]. The parameters of the ARMA model are determined using a grid search and its BIC score. We then compare the root-mean-squared-error between samples from the predictive distribution obtained by BNDD and the true post-intervention signal, and similarly evaluate the performance of the ARMA extrapolations and the true signal. An example simulation run and BNDD application is shown in Fig 3A, with a post-intervention frequency shift of $\alpha = 4$Hz. The model correctly recovers the true power spectrum, as well as the decreased amplitude of the second harmonic component post-intervention, and finds barely worth mentioning evidence in favor of an effect (logBF = 0.15). The estimated spectral mixture of the continuous model is centered between the true frequencies of the control and intervention group (Fig 3B). This faithfully represents the null hypothesis that the observations can be explained without any changes in spectral content. As the discontinuity grows larger, the standard deviation of the components of the continuous model increases as well, since it has to account for a larger difference. $\mathcal{M}_1$ instead correctly identifies the true mixture components. Fig 3C shows the RMSE of samples from the posterior distributions of $f_1$ and the true function, as well as the ARMA estimate. BNDD consistently outperforms the baseline.

## Applications

### The effects of winning an election and longevity

A recent study [44] investigated the effect of running for US gubernatorial office on longevity. The authors use a regression discontinuity design, and conclude that politicians winning a close election live 5 to 10 years longer than if they had lost. These findings have been heavily criticized [45], and it is unclear whether a regression discontinuity analysis is actually

appropriate here, as there is no clear intervention at $x = 0$ (where $x$ is the percentile difference in election result). Despite these concerns, we analyze this data set here as it allows us to demonstrate some of the functionality specific to BNDD. The data are available from the original publication [44], and subsequently preprocessed following [45].

Using the linear RDD baseline we find an optimal bandwidth of 5.48 percentile points using the Imbens-Kalyanaraman procedure [30]. When using this bandwidth and testing for an effect, we find $p = 0.019$ and an estimated effect size of 9.4 years. With BNDD, using either a linear, exponential, or Matérn ($v = 3/2$) covariance function, we find a more parsimonious explanation of these data by a constant function and a substantial noise term $\sigma_n^2$, as shown by log Bayes factors of -0.12, 0.0, and 0.0, respectively, as shown in Fig 4. This indicates that from these data, no clear conclusion can be drawn, and that such a scenario is clearly identified using BNDD.

## Phantom border effect on Dutch government elections

In 2017, the Dutch general elections were held. According to Dutch electorate geographer De Voogd, the share of votes that go to populist parties (We refrain from an extensive discussion of the definition of populism and refer to populist parties as those parties that emphasize 'an alleged chasm between the elite and the general population'. In the Dutch 2017 elections, parties that fit this description were PVV, SP, 50Plus and FvD [46].) is different north and south of a so-called 'phantom border', a line that historically divided the catholic south of the Netherlands from the protestant north [47, 48]. This border serves as a two-dimensional threshold along which one can apply RD design. This special case of RD design where the assignment threshold is a geographical boundary is also referred to as GeoRDD [10]. Here, we test the hypothesis by De Voogd.

First, the vote distribution per Dutch municipality were collected from the Dutch government website [49]. We then manually constructed an approximation of the phantom border (see the dashed lines in Fig 5) and used this as a function to divide the available municipalities in either above or below the border. For visualization of country and municipality borders, data from the Dutch national georegister was used [50]. Next, we applied BNDD using the linear and first-order Matérn covariance functions. The results of the analysis are shown in Fig 5. The figure shows the Netherlands with the fraction of populist votes per municipality

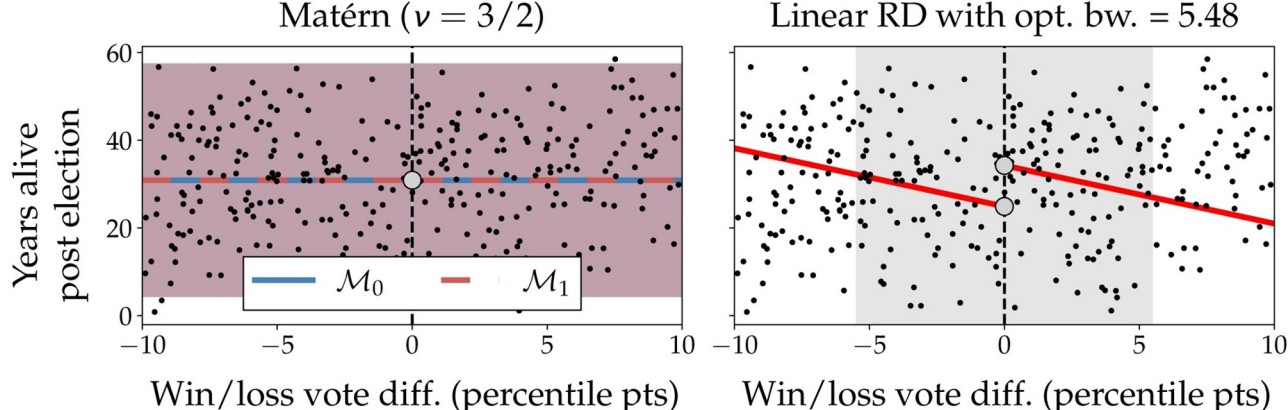

**Fig 4. Election result effects on longevity.** Discontinuity analysis of the effect of close gubernatorial elections on longevity [44]. Shown are regressions by BNDD using a Matérn covariance function, and a linear RD baseline with an optimized bandwidth of 5.48 percentile points (shaded area). For BNDD, the regressions for $\mathcal{M}_0$ and $\mathcal{M}_1$ are nearly identical. For the baseline, the bandwidth optimization leads to a poor linear fit, and hence a spurious detection of an effect.

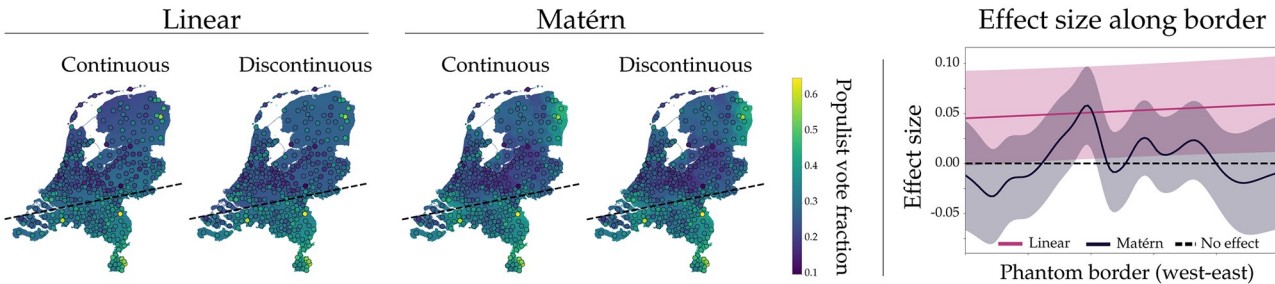

**Fig 5. Phantom-border effects on populist voting.** Discontinuity analysis along a two-dimensional boundary (indicated by the dashed line). **A**. Circles indicate the observed fraction of populist votes; municipalities are shaded according to the Gaussian process predictions. **B**. The distribution of effect size conditioned on $\mathcal{M}_1$, $p(d \mid D, \mathcal{M}_1)$, along the phantom border. The shaded interval indicates one standard deviation around the mean. The country and municipality border data are available at the website of the Dutch national georegister [50], and the superimposed populist voting fractions were derived from the 2017 election results at https://data.overheid.nl [49].

superimposed, together with the phantom border representing the supposed divide in voting behaviour.

If we assume a linear underlying process, there is strong evidence for a discontinuity (logBF = 24.4), confirming the hypothesis by De Voogd. Visually however, the data do not appear to follow these linear trends. The nonparametric Matérn covariance function results in evidence *against* an effect (logBF = −3.5). As the Matérn covariance function fits the data much more accurately than the linear covariance function, the Bayesian model average is completely dominated by the former, leading to the conclusion that the historical phantom border does not create a geographic discontinuity in populist voting behaviour.

## Kundalini meditation effect on heart rate

Earlier work [51] studied the hypothesis that Kundalini Yoga meditation techniques reduce one's heart rate. However, they find the opposite; the meditation instead brings about an increase in heart rate. The experiment lends itself well for ITS design, but in practice may be difficult to perform because the data are not evenly sampled. However, this is not a prerequisite for Gaussian process regression, which together with the spectral mixture kernel [15] is well-suited to model these data. The observations are obtained from the PhysioNet database and consists of heart rates of two women and two men, of ages 20–52 (mean 33) [52]. We focus on one participant due to space constraints. Since we do not merely want to detect a change in absolute heart rate, but in its fluctuations, use a changepoint mean function [53] for $\mathcal{M}_0$ and two separate constant mean functions for $\mathcal{M}_1$ to capture the different means. Fig 6 shows the corresponding regression and extrapolation. The continuous model requires more spectral mixture components; $Q = 6$ for $f_0$ compared to $Q = 2$ for $f(x; \theta_1^{\mathbf{A}})$ and $Q = 3$ for $f(x; \theta_1^{\mathbf{B}})$. The analysis finds overwhelming evidence for an effect (logBF = 281.2).

## Discussion

In order to infer causality from QED, one assumes that the alleged change occurs at the threshold, but that the latent process is otherwise stationary. Consequently, the behaviour of the two groups changes sharply around the intervention. In standard RD studies, this locality is controlled via a bandwidth parameter that determines the sensitivity of the detection approach [20]. This requires the availability of sufficient data around the threshold, and the analysis is sensitive to this parameter. In BNDD with stationary nonparametric covariance functions, the bandwidth is replaced by a length-scale hyperparameter, which we optimize using the model

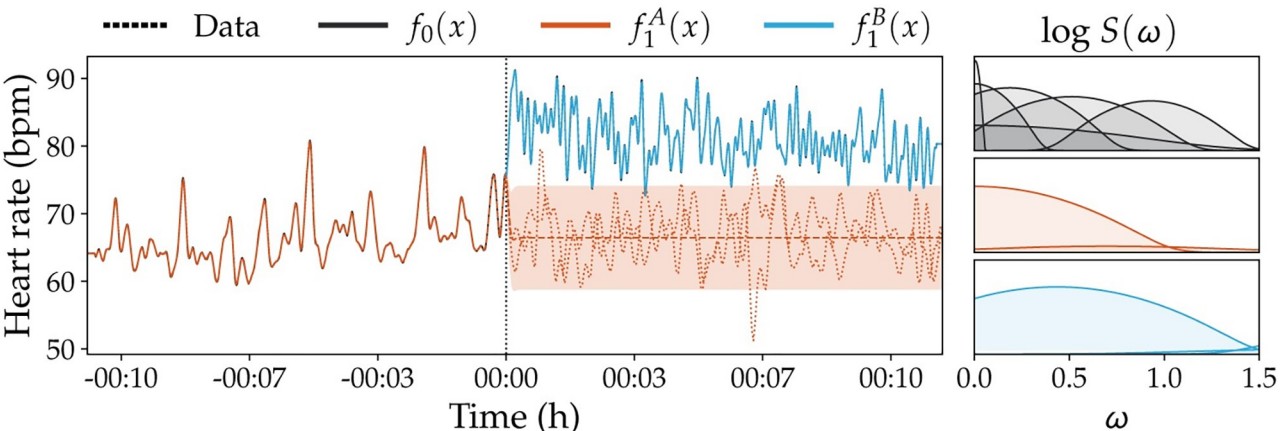

**Fig 6. Kundalini meditation effect on heart rate.** Analysis of meditation effect on heart rate. Shown is the participant's heart rate, who starts meditation at $x_0 = 00:00$. The extrapolation, indicated by the dashed (mean) and dotted (posterior samples) red lines, is poor in comparison to the actual observations, which is corroborated by the large log Bayes factor. The panel on the right shows the (log) power spectra expressed by the optimized covariance function hyperparameters.

marginal likelihood. The length-scale regulates how fast the correlations between consecutive points decay with their distance, and thus how sensitive BNDD is around the threshold [54]. This implements a trade-off between estimation reliability and the locality assumption that is needed to draw causal inferences [16]. The benefit of a length-scale instead of a fixed bandwidth parameter is that the relative influence of observations decreases gradually as they are further away from the intervention point, and that this distance is automatically adjusted.

With an exponential covariance function the most rigorous form of locality can be enforced. Here, the Markov properties of the Gaussian process guarantee that only discontinuities at the intervention threshold are detectable. On the other hand, non-local covariance functions such as the periodic covariance function are vulnerable to false positives if the true process is non-stationary. Here, the presence of change points away from the intervention threshold can lead to false alarms, due to the flexibility of the regressions. In this case, or in exploratory applications, BNDD can be performed in a sliding-window fashion to ensure that the highest Bayes factor is at the intervention threshold.

The Bayesian model averaging procedure that we use in BNDD depends on the model probabilities $p(\mathcal{M}_0)$ and $p(\mathcal{M}_1)$. Here, we have assumed a uniform prior on these model probabilities, as we have no reason to prefer either the continuous null model or the discontinuous alternative. However, it should be noted that prior beliefs may be incorporated to reflect our initial assumptions on the probability of an effect, as well as to adjust for multiplicity in case many hypotheses are tested simultaneously [36] (for instance, in [55] a regression discontinuity design is used to test the causal influence between neuronal populations).

BNDD can be extended in several ways. For instance, we do not currently account for covariates that may serve as confounds for causal inference [19, 35]. However, such covariates can be explicitly taken into account in the regression models, or even be learned from the observations [56]. Covariate selection can be performed using automatic relevance determination [57], where we learn separate length-scales for each covariate. Furthermore, improvement is expected from more accurate estimators of the model marginal likelihood than the BIC, such as the ELBO or bridge sampling [58]. Throughout this paper, we have assumed a Gaussian likelihood. This conveniently leads to an analytic solution of the GP posterior, because the GP prior is conjugate to this likelihood. However, using variational inference or the Laplace

approximation, BNDD can be used in combination with non-Gaussian observation models [37]. For instance, one could use a Poisson likelihood to model observed count data [59], or a Bernoulli likelihood for binary observations [60]. Furthermore, other nonparametric priors over the latent functions may be used, such as the Student t- process [61]. Other extensions include modelling nondeterministic application of the threshold assignment, delayed response functions, multi-dimensional response variables [62]. BNDD extends naturally to the setting of multiple assignment variables [18, 63–66].

## Conclusion

In all, BNDD serves as a Bayesian nonparametric approach for causal inference in quasi-experimental designs. By selecting the appropriate covariance function, one has precise control over the type of discontinuity that can be detected, as well as a priori assumptions of the latent data generating processes. Importantly, Bayesian model averaging allows us to marginalize over key assumptions, such as the choice of covariance function, or the presence/absence of an effect. The resulting method is a nuanced framework for discontinuity-based research designs.

## Author Contributions

**Conceptualization:** Max Hinne, David Leeftink, Marcel A. J. van Gerven, Luca Ambrogioni.

**Data curation:** Max Hinne, David Leeftink.

**Formal analysis:** Max Hinne, David Leeftink, Luca Ambrogioni.

**Investigation:** Max Hinne, David Leeftink, Luca Ambrogioni.

**Methodology:** Max Hinne.

**Software:** Max Hinne, David Leeftink.

**Supervision:** Max Hinne, Marcel A. J. van Gerven, Luca Ambrogioni.

**Validation:** Max Hinne.

**Visualization:** Max Hinne.

**Writing – original draft:** Max Hinne.

**Writing – review & editing:** Max Hinne, David Leeftink, Luca Ambrogioni.

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
