## [Decision Letter · Decision Letter 0]

27 Apr 2022

PONE-D-22-03933Bayesian model averaging for nonparametric discontinuity designPLOS ONE

Dear Dr. Hinne,

Thank you for submitting your manuscript to PLOS ONE. After careful consideration, we feel that it has merit but does not fully meet PLOS ONE’s publication criteria as it currently stands. Therefore, we invite you to submit a revised version of the manuscript that addresses the points raised during the review process.

The authors must address all the issues raised by the reviewer. The paper is recommended for major revision.

We look forward to receiving your revised manuscript.

Kind regards,

Lalit Chandra Saikia, PhD

Academic Editor

PLOS ONE

Journal Requirements:

- https://arxiv.org/abs/1911.06722

In your revision ensure you cite all your sources (including your own works), and quote or rephrase any duplicated text outside the methods section. Further consideration is dependent on these concerns being addressed.

3. We note that Figure 5 in your submission contain map images which may be copyrighted. All PLOS content is published under the Creative Commons Attribution License (CC BY 4.0), which means that the manuscript, images, and Supporting Information files will be freely available online, and any third party is permitted to access, download, copy, distribute, and use these materials in any way, even commercially, with proper attribution. For these reasons, we cannot publish previously copyrighted maps or satellite images created using proprietary data, such as Google software (Google Maps, Street View, and Earth). For more information, see our copyright guidelines: http://journals.plos.org/plosone/s/licenses-and-copyright.

 a. You may seek permission from the original copyright holder of Figure 5 to publish the content specifically under the CC BY 4.0 license. 

Additional Editor Comments:

The authors must address all the issues raised by the reviewer. The paper is recommended for major revision.

Reviewers' comments:

Reviewer's Responses to Questions

**Comments to the Author**

1. Is the manuscript technically sound, and do the data support the conclusions?

Reviewer #1: Yes

Reviewer #2: Yes

2. Has the statistical analysis been performed appropriately and rigorously? 

Reviewer #1: Yes

Reviewer #2: Yes

3. Have the authors made all data underlying the findings in their manuscript fully available?

Reviewer #1: Yes

Reviewer #2: Yes

4. Is the manuscript presented in an intelligible fashion and written in standard English?

Reviewer #1: Yes

Reviewer #2: Yes

5. Review Comments to the Author

Reviewer #1: The paper presents a framework for discontinuity-based designs using Bayesian model averaging and Gaussian process regression. It is a topic of interest to the researchers in the related areas. For the reader, however, a number of points need clarifying and certain statements require further justification. My detailed comments are as follows:

(1)In the abstract section, the quantification proposed by the authors supporting the innovation point of either model is crucial to solve the main problem of the average of Bayesian models, because the prior probabilities of different models greatly affect the results of the Bayesian model averaging method, and the authors please describe them in detail in this section.

(2)Is this article more applicable to the discontinuity model?The study of continuous models is not comprehensive and detailed, does this paper focus on the description of discontinuous models?

(3)It is noteworthy that your paper requires careful editing of the format.There are many problems in the paper format, such as the first line of the paragraph, multiple syntax errors, etc.

Reviewer #2: This paper is quite interesting paper working on nonparametric discontinuity design with Bayesian model averaging. Previous approaches have overconfidence from the implicit consumptions and model misspecification from the overly simplified regression models. To overcome these shortcomings, this paper proposes a framework of discontinuity-based designs using Bayesian model averaging and Gaussian process regression, namely, Bayesian nonparametric discontinuity design' (BNDD).

This paper demonstrated the usage of BNDD in multiple promising simulations, such as the effects of winning an election and longevity, phantom border effect on Dutch government elections, and Kundalini meditation effect on heart rate.

The paper is also well written and structured, and it is easy to follow.

One minor doubt for me is the paper assume the distribution is Gaussian distribution, and thus Gaussian process is used in the paper. Is possible real world simulation is not under Gaussian distribution, and thus can not be used in Gaussian process?

6. PLOS authors have the option to publish the peer review history of their article (what does this mean?). If published, this will include your full peer review and any attached files.

Reviewer #1: No

Reviewer #2: No

---

## [Author Response · Author response to Decision Letter 0]

23 May 2022

Please see the included 'Response to Reviewers.pdf' file for a detailed response.

---

## [Decision Letter · Decision Letter 1]

8 Jun 2022

Bayesian model averaging for nonparametric discontinuity design

PONE-D-22-03933R1

Dear Dr. Hinne,

We’re pleased to inform you that your manuscript has been judged scientifically suitable for publication and will be formally accepted for publication once it meets all outstanding technical requirements.

Kind regards,

Lalit Chandra Saikia, PhD

Academic Editor

PLOS ONE

Additional Editor Comments (optional):

Reviewers' comments:

Reviewer's Responses to Questions

**Comments to the Author**

1. If the authors have adequately addressed your comments raised in a previous round of review and you feel that this manuscript is now acceptable for publication, you may indicate that here to bypass the “Comments to the Author” section, enter your conflict of interest statement in the “Confidential to Editor” section, and submit your "Accept" recommendation.

Reviewer #1: All comments have been addressed

2. Is the manuscript technically sound, and do the data support the conclusions?

Reviewer #1: Yes

3. Has the statistical analysis been performed appropriately and rigorously? 

Reviewer #1: Yes

4. Have the authors made all data underlying the findings in their manuscript fully available?

Reviewer #1: Yes

5. Is the manuscript presented in an intelligible fashion and written in standard English?

Reviewer #1: Yes

6. Review Comments to the Author

Reviewer #1: This paper can be accepted, the research in this paper is interesting, and the authors have done enough work.

7. PLOS authors have the option to publish the peer review history of their article (what does this mean?). If published, this will include your full peer review and any attached files.

Reviewer #1: No

---

## [Editor Report · Acceptance letter]

14 Jun 2022

PONE-D-22-03933R1 

Bayesian model averaging for nonparametric discontinuity design 

Dear Dr. Hinne:

I'm pleased to inform you that your manuscript has been deemed suitable for publication in PLOS ONE. Congratulations! Your manuscript is now with our production department. 

Kind regards, 

on behalf of

Dr. Lalit Chandra Saikia 

Academic Editor

PLOS ONE